# Inhibitors of Chemoresistance Pathways in Combination with Ara-C to Overcome Multidrug Resistance in AML. A Mini Review

**DOI:** 10.3390/ijms22094955

**Published:** 2021-05-07

**Authors:** Guadalupe Rosario Fajardo-Orduña, Edgar Ledesma-Martínez, Itzen Aguiñiga-Sánchez, María de Lourdes Mora-García, Benny Weiss-Steider, Edelmiro Santiago-Osorio

**Affiliations:** 1Hematopoiesis and Leukemia Laboratory, Research Unit on Cell Differentiation and Cancer, FES Zaragoza, National Autonomous University of Mexico, 09230 Mexico City, Mexico; guadalupefajardo@hotmail.com (G.R.F.-O.); 2814.260@gmail.com (E.L.-M.); liberitzen@yahoo.com.mx (I.A.-S.); bennyweiss@hotmail.com (B.W.-S.); 2Department of Biomedical Sciences, School of Medicine, Faculty of High Studies Zaragoza, National Autonomous University of Mexico, 09230 Mexico City, Mexico; 3Immunobiology Laboratory, Research Unit on Cell Differentiation and Cancer, FES Zaragoza, National Autonomous University of Mexico, 09230 Mexico City, Mexico; mogl@unam.mx

**Keywords:** leukemia, relapse, refractory disease, drug resistance, overcoming chemoresistance

## Abstract

Acute myeloid leukemia (AML), the most common type of leukemia in older adults, is a heterogeneous disease that originates from the clonal expansion of undifferentiated hematopoietic progenitor cells. These cells present a remarkable variety of genes and proteins with altered expression and function. Despite significant advances in understanding the molecular panorama of AML and the development of therapies that target mutations, survival has not improved significantly, and the therapy standard is still based on highly toxic chemotherapy, which includes cytarabine (Ara-C) and allogeneic hematopoietic cell transplantation. Approximately 60% of AML patients respond favorably to these treatments and go into complete remission; however, most eventually relapse, develop refractory disease or chemoresistance, and do not survive for more than five years. Therefore, drug resistance that initially occurs in leukemic cells (primary resistance) or that develops during or after treatment (acquired resistance) has become the main obstacle to AML treatment. In this work, the main molecules responsible for generating chemoresistance to Ara-C in AML are discussed, as well as some of the newer strategies to overcome it, such as the inclusion of molecules that can induce synergistic cytotoxicity with Ara-C (MNKI-8e, emodin, metformin and niclosamide), subtoxic concentrations of chemotherapy (PD0332991), and potently antineoplastic treatments that do not damage nonmalignant cells (heteronemin or hydroxyurea + azidothymidine).

## 1. Introduction

Acute myeloid leukemia (AML) is a hematological disease characterized by the clonal expansion of immature myeloid hematopoietic cells (myeloblasts), which accumulate in the bone marrow (BM) and blood (more than 20% of myeloblasts) and can also infiltrate tissues such as the spleen, liver, skin, gums, and central nervous system. The displacement of functional hematopoietic cells in the bone marrow causes marrow failure with a consequent deficit in the production of the cellular components of blood: erythrocytes, leukocytes and platelets [1,2,3]. For the past 50 years, the standard of care in AML has been the combination of cytarabine (Ara-C) and anthracyclines, resulting in complete remission of no more than 45% of patients regardless of age group [4]; however, most of them do not survive long-term (more than five years), largely due to treatment-related mortality and the appearance of relapse associated with chemoresistance [2,5,6]. In this sense, drug resistance that initially occurs in leukemic cells or that develops during or after treatment is one of the most difficult challenges in AML therapy. In recent years, a greater understanding of the molecular biology of leukemia has allowed the development of therapies directed towards mutated genes and molecular targets; however, the new treatments approved by the US Food and Drug Administration (FDA) are only effective in some patients, and the survival period that is achieved does not differ from that obtained with standard therapy. It is clear that more effective and less toxic strategies are required [1,7]. However, the characterization of AML as a highly heterogeneous disease involving a great variety of genes and proteins with altered expression or function simultaneously explains, at least in part, why therapies directed against a single molecular target may not be sufficient to increase the survival of patients with AML. Since 2017, several agents have been approved for various indications in AML; thus, improvement is expected with these new treatment modalities, but understanding and modifying resistance mechanisms to currently available drugs may be another useful way to manage this disease. In this work, the main molecules related to the induction of drug resistance and the different therapeutic strategies that seek to stop this condition are addressed, particularly those that favor the sensitization of leukemic cells to Ara-C, the main drug used in the standard treatment of AML.

## 2. Acute Myeloid Leukemia (AML) and Its Treatment

AML, the most common type of leukemia in older adults, is a heterogeneous disease that originates from the clonal expansion of undifferentiated hematopoietic progenitor cells. AML is slightly more common in men than in women and less frequently affects children and young people [3]. The median age at diagnosis is 65 years, and the incidence increases progressively with age, making it the most common type of leukemia in older people [1,6]. Due to the heterogeneity of the disease, different classifications of AML subtypes have been proposed. The French–American–British (FAB) classification was the first and proposes eight subtypes of AML according to the morphological characteristics of the cells and their maturity [3]. However, even though the FAB classification is still used to group patients into different subtypes, diagnosis and treatment are determined by the World Health Organization (WHO) and European LeukemiaNet (ELN) classification systems. In 2017, the WHO carried out a review and updated its AML subtype classification based on the morphological, cytochemical and immunophenotypic characteristics and on gene sequencing and expression data. The updated classification includes six categories: AML with recurrent genetic abnormalities, AML with myelodysplasia-related changes, therapy-related myeloid dysplasia, AML not otherwise specified, myeloid sarcoma and Down syndrome-related myeloid proliferation [8]. The ELN classification proposes three groups: favorable risk, intermediate risk and adverse risk, which are primarily based on pretreatment cytogenetic abnormalities in the NPM1, FLT3, CEBPA, RUNX1, ASXL and TP53 genes [2]. Short et al. proposed a simplified classification of AML based on the sensitivity and resistance to chemotherapy at the time of diagnosis (Table 1). Considering the large impact that the development of chemoresistance has on the development of relapse and refractory disease, it is reasonable that it is considered in the classification of AML [9,10].

The development of additional therapies focused on targeted genomic, molecular or cellular subgroups aims to create drugs that are more effective and less toxic [6]. Such therapies include inhibitors of molecular aberrations, pro-apoptotic agents, microenvironment-target molecules, cell cycle checkpoint inhibitors, epigenetic regulators, monoclonal and bispecific antibodies, and cells with a chimeric-specific receptor (CAR cells), among others [11,14,15,16,17]. The clinical activity of some of these drugs has been described as “promising”, while others have been approved by the FDA or are in clinical phases I, II or III trials (Table 2). Unfortunately, the therapeutic results have been unsatisfactory, as evidenced by the AML mortality rate, which did not significantly change throughout 2005–2016. This contrasts with an appreciable decrease in the mortality rates of patients with chronic lymphocytic leukemia (CLL) or chronic myeloid leukemia (CML) during the same period [18,19]. The standard antineoplastic therapy for more than 50 years continues to include an induction stage followed by a consolidation stage. The first stage corresponds to the intensive or 7 + 3 scheme, which consists of seven days of administration of the cytokine arabinoside antimetabolite (1-β-arabinofuranosylcytosine “Ara-C”) or cytarabine, followed by three days of anthracycline (daunorubicin or idarubicin) [2]. After this first cycle of induction therapy, between 25% and 50% of patients still have cytological traces of the disease. These patients should receive a second cycle of the 7 + 3 scheme, high doses of Ara-C alone, or a combination of cytarabine with fludarabine and filgrastim (FLAG) plus idarubicin (FLAG-IDA). Subsequently, the consolidation stage of therapy aims to prevent relapse and eradicate minimal residual disease, which is based on Ara-C or on allogeneic hematopoietic cell transplantation (HCT). According to the National Comprehensive Cancer Network (NCCN) and ELN clinical practice guidelines, for patients younger than 60 years with a recent diagnosis, the standard doses of induction treatment are recommended, while for patients older than 60 years, treatment is decided based on the risk group and functional status [3].

## 3. Chemoresistance as Relapse and Refractory Disease

In general, patients with a recent diagnosis of AML in the absence of treatment have a survival of days or weeks. With standard therapy, between 60% and 75% of children achieve an overall survival of 5 years, largely due to fewer genetic mutations and their greater ability to tolerate both high-intensity chemotherapy and HCT [1,20,21]. In contrast, only 24% of adult patients who are selected for intensive therapy and 10%–15% of older patients not eligible for intensive therapy survive to five years, [1,6,22,23] and almost 80% of patients diagnosed at an age ≥65 years die within one year [19]. AML in the elderly population is a special therapeutic challenge due to the poor functional status of this age group, which results in an increase in therapy-related toxicity [24]. Additionally, this group of patients also presents greater resistance to treatment [25] due to overexpression of the MDR1 multidrug resistance gene [26], and the development of chemoresistance related to an adverse karyotype is more frequent [6].

Relapse and refractory disease (R/R) are major problems that prevent an adequate response to AML treatment. Because relapse refers to the reappearance of disease, it is widely related to refractory disease, which in turn leads to a poor prognosis [27]. Refractory disease is defined as any failure to achieve complete remission after two cycles of induction therapy [12,28]. R/R may exist prior to exposure to chemotherapeutic agents (intrinsic or primary resistance), or it can develop or increase during drug treatment (acquired or secondary resistance), leading to chemoresistance and relapse [29]. Sixty percent of patients in the favorable risk category and 85% in the adverse risk category develop R/R, and only 10–20% survive to five years [13,14,15,16,17,18,19,20,21,22,23,24,25,26,27,28,29,30].

Analyzing samples from patients with AML at diagnosis and at relapse can reveal differences in the mutational profile before and after treatment. Mutations in genes associated with signaling activation are commonly found at diagnosis, but these are frequently undetected at the time of relapse, suggesting that these mutations have a greater role in the development of leukemia and are less important for chemotherapy avoidance [27]. Furthermore, approximately 30%–40% of patients show cytogenetic alterations at the time of relapse that were not found at the initial diagnosis, which suggests a clonal evolution of the disease [30]. The transformation of AML cells from a sensitive state to one resistant to treatment, due to exposure to a chemotherapeutic agent, can favor the development of cross-resistance to other structurally unrelated components, resulting in a phenotype called multidrug resistance (MDR) [29]. MDR is a multifactorial phenomenon explained by a variety of molecular mechanisms of chemoresistance (MOC) that include several genes or proteins with altered expression or function, leading to a reduction in the efficacy of anticancer agents against target mechanisms (Figure 1).

One group of MOCs includes genes that affect the intracellular amount of the active drug. These MOCs can reduce drug entry into the cytoplasm of leukemic cells by hindering either the expression, function or both, of the solute carriers responsible for drug absorption, or they can increase the ability of tumor cells to export pharmacologically active agents, which are exported by pumps or efflux transporters belonging to the ATP-binding cassette superfamily (ABCC4, ABCC10 and ABCC11). Another group of MOCs can lead to a decreased ratio of active versus inactive agents within tumor cells due to changes in either the expression, function or both, of enzymes responsible for prodrug activation or drug inactivation [29,31,32]. Other mechanisms that affect the response to chemotherapy include changes in the expression or function of the molecular targets of anticancer drugs; an increase in the ability of cancer cells to repair the DNA damage induced by anticancer drugs; either decreased expression, function or both of proapoptotic factors; upregulation of antiapoptotic genes; and increased responses to stress and changes in the leukemic cellular microenvironment [29]. microRNAs (miRNAs), which are the most studied noncoding RNAs (ncRNAs) in cancer, also represent a chemoresistance mechanism in AML [33,34,35]. In the case of AML, two groups of miRNAs related to resistance to intensive treatments have been determined: group I includes miRNAs in which increased expression leads to sensitivity or resistance to the drug, and group II includes miRNAs in which increased expression favors resistance while a decrease leads to sensitivity [33] (Table 3).

Likewise, other mechanisms of drug resistance based on the regulation of individual genes in AML relapse have been proposed; these genes include SAMHD1 (SAM and HD domain containing deoxynucleoside triphosphate triphosphohydrolase 1), EZH2 (enhancer of zeste homolog 2) and KDM6A (lysine demethylase 6A). SAMHD1 encodes a deoxynucleoside triphosphate triphosphohydrolase. In AML, SAMHD1 expression is inversely correlated with the response to Ara-C in vitro and in vivo; thus, the SAMHD1 expression level is considered a predictive marker of response to Ara-C. EZH2 encodes a histone methyltransferase and is commonly mutated at the time of AML diagnosis. In relapse, its inactivation is associated with a poor prognosis, relapse and drug resistance. KDM6A mutations are recurrent in relapse and in vitro, and their inactivation leads to an increase in resistance to Ara-C and daunorubicin, while their re-expression sensitizes cells to treatment [17,27,36,38,39].

Other important factors that contribute to the development of chemoresistance are leukemic stem cells (LSCs) and their aberrant oxygen metabolism. The concept of LSCs has been established to explain subclonal architectures and hierarchies in leukemias. LSCs have an unlimited capacity to divide and self-renew, thereby propagating the malignancy for unlimited time periods [40]. Chemotherapy resistance in cancer stem cells (CSCs) can be attributed to their reduced proliferation or quiescence, heightened DNA repair, reduced apoptosis, increased clearance of reactive oxygen species (ROS), enhanced drug efflux mechanisms and reduced immune clearance [41]. However, perhaps the most compelling phenotype separating CSCs from normal stem cells is aberrations in the dependencies on oxygen and lipid metabolism. A previous study reported the dependency of LSCs and therapy-resistant AML cells on oxidative metabolism in AML patients [42]. BCL-2 is an anti-apoptotic member of the BCL-2 protein family and is overexpressed in LSCs, suggesting that targeting BCL-2 can result in the preferential elimination of this cell population. Ongoing clinical trials are investigating the efficacy of the bioavailable BCL-2 inhibitor venetoclax, in combination with the hypomethylating agent azacytidine, to therapeutically target primitive LSCs, which was previously shown in vitro and in vivo [43,44]. Treating older AML patients with venetoclax in combination with azacytidine resulted in remissions that were superior to conventional treatment regimens. An analysis of LSCs from these patients showed disruption of the TCA cycle manifested by decreased alpha-ketoglutarate and increased succinate levels, suggesting inhibition of electron transport chain complex II. In vitro modeling confirmed the inhibition of complex II via reduced glutathionylation of succinate dehydrogenase. These metabolic perturbations suppress oxidative phosphorylation (OXPHOS), which targets LSCs. Aberrant oxygen metabolism is an MOC and an option to eradicate LSCs in AML patients by disrupting the metabolic machinery driving energy metabolism [44].

Knowing the different MOCs in each patient at each moment of their AML clinical history, that is, before, during and after treatment and relapse, would enable the use of personalized therapies with better results and less toxicity, which would result in a longer survival time. Some authors suggest that achieving this enterprise faces issues such as study costs and the technical limitations of molecular testing [45], others are more optimistic, recently a bone marrow mononuclear cell (BMMC) 3D culture model and an automated flow cytometry method, were tested to evaluating the sensitivity of leukemia cells to multiple chemotherapeutic drugs ex vivo and define resistance biomarkers of AML patient bone marrow samples in response to Ara-C treatment [46,47]. In silico analysis of highly prognostic human AML LSC gene expression signatures using existing datasets of drug-gene interactions have been used to identify compounds predicted to target LSC gene programs [48], the identification of a true genetic signature that predicts the response to cytarabine has been assayed in patients with AML through comprehensive multi-point analysis pharmacokinetic, pharmacodynamic (cytarabine-dependent) and clinical endings [49]. In this regard, a comprehensive understanding of possible prognostic biomarkers from preclinical and clinical studies of chemoresistance in AML is available, in addition to delineating the most prevalent mutations associated with chemoresistance mechanisms and to outlining future directions to improve current prognosis tools, in order to have a guide that will facilitate therapeutic decision making towards a personalized prognosis and improved therapeutic efficacy [17]. As has been proposed in several reports (Table 4), here we describe that the key to increasing AML patient survival might be the chemosensitization of leukemic cells to chemotherapy with Ara-C, the most successful antineoplastic drug for the treatment of AML. It is necessary to design strategies that avoid the cellular exclusion of the drug, a common mechanism in the induction of resistance. These strategies should also simultaneously modulate different key signaling pathways that are known to determine both leukemogenesis and the appearance of chemoresistance [9,29,50] and that are associated with metabolic dysregulation, the expression of transcription factors and oncogenes, the suppression of tumors and apoptosis, cell cycle progression and angiogenesis. Next, we discuss the evidence indicating that chemosensitizing AML cells to Ara-C is possible without causing irreparable damage to normal tissue.

## 4. AML Cell Chemosensitization to Ara-C

For Ara-C to function as an antitumor agent, it must first be transported into leukemic cells at high doses (plasma levels greater than 10 µM). This agent can penetrate the plasma membrane by simple diffusion [67]; otherwise, as occurs in patients who are not candidates for the intensive chemotherapy regimen, Ara-C requires membrane transporters, mainly human balancing nucleoside transporter 1 (hENT1) [68]. Once inside the cell, Ara-C is phosphorylated by deoxycytidine kinase (dCK) to Ara-C monophosphate (Ara-CMP) and then by deoxycytidine monophosphate kinase (dCMP) to Ara-C diphosphate (Ara-CDP). Finally, nucleoside diphosphate kinase (NDPK) produces Ara-C triphosphate (Ara-CTP), the active metabolite of Ara-C that is incorporated into DNA during the S phase and blocks cell cycle progression, eventually leading to apoptotic cell death (Figure 2a). Since the cytotoxicity of Ara-C depends on the incorporation of Ara-CTP into DNA, the intracellular levels of Ara-CTP represent an index of the drug’s cytotoxicity; the intracellular pharmacokinetics of Ara-CTP are correlated with the response to therapy [67,68]. The primary mechanisms underlying Ara-C resistance appear to be the intracellular levels of the active metabolite Ara-CTP, which may be due to low levels or low activity of the hENT1 transporter, reduced levels of activating enzymes (mainly dCK), increased levels of inactivating enzymes (CDA, NT5C2, DCTD and SAMHD1), or a cellular increase in the dCTP pool, which can compete with the incorporation of Ara-CTP into DNA and inhibit dCK activity through feedback inhibition, thus preventing the production of the active form (Figure 2b; Table 3) [69].

Previous studies have shown that administering the purine nucleotide analog fludarabine in conjunction with Ara-C considerably increases the accumulation rate of intracellular Ara-CTP in leukemic blasts [70]. Based on these findings, this ribonucleotide reductase inhibitor has been administered together with Ara-C and has functioned as an interesting and effective antileukemic regimen in R/R cases [71,72]. This approach has been expanded for several other drugs [73,74,75,76] and has appeared to be relatively effective and even superior to standard regimens in some phase I/II trials [77]. Most of these studies agree that more clinical trials are needed before considering a change to the AML treatment paradigm. However, the evidence suggests that the synergy strategy works, and it is likely that not just one but multiple drugs administered together with Ara-C is the key to overcoming resistance and relapse in AML.

### 4.1. Synergistic Cytotoxicity with Ara-C

The factors that reduce the amount of Ara-CTP are known to induce resistance to Ara-C; however, not all mechanisms of Ara-C resistance can be explained by these factors [67,68]. In vitro studies have shown that intracellular concentrations of Ara-CTP are higher in Ara-C-sensitive cells than in Ara-C-resistant cells. Likewise, cells from patients with AML that do not respond to Ara-C have only half the levels of Ara-CTP compared to patients who do respond to the drug [69]. Likewise, the efflux pumps ABCC4 (MRP4), ABCC10 (MRP7) and ABCC11 (MRP8) have been linked to Ara-C resistance because they are transporters of this drug and can decrease its concentration in AML cells (Figure 3a). One strategy to limit the ability of cells to exclude Ara-C and its monophosphorylated form (Ara-CMP) is to inhibit the functioning of ABCC4, a transmembrane protein belonging to the ATP-binding cassette transporter superfamily (ABC) and highly expressed in AML [31,37,51,67,78]. ABCC4 inhibition can be accomplished with the use of inhibitors such as sorafenib or MK571 or by silencing it with siRNA (Figure 3b). Unfortunately, this strategy also sensitizes parental myeloids to Ara-C; therefore, it also induces toxicity and exacerbates hematological damage in healthy cells [51,79,80,81,82,83]. This is a serious drawback and remains a challenge to be addressed.

Avoiding drug exclusion is the first step for an effective therapy. Leukemic cells must be sensitized to chemotherapy drugs before they can develop resistance, as well as when they are already chemoresistant during disease relapse. Next, we describe the induction of synergistic cytotoxicity in chemoresistant leukemic cells by combining Ara-C with other molecules (MnKI-8e, emodin, metformin and niclosamide), in reduced concentrations of Ara-C (PD0332991) and in combination strategies (heteronemin, hydroxyurea, and azidothymidine) might limit damage to nonmalignant cells.

Prolonged exposure to Ara-C causes a positive regulation of several survival molecules of the mitogen-activated protein kinase (MAPK) pathway. MAPK-interacting kinases (Mnks) are related to the expression of antiapoptotic proteins (Mcl-1 and Bcl-2) [84], oncoproteins (c-Myc and cyclin D1) (Figure 3a) and inducers of angiogenesis (VEGF). It has been proposed that inhibiting these molecules could sensitize leukemia cells that are resistant to chemotherapy. Because Mnks do not seem to have an essential role in normal development, the inhibition of these molecules could provide an effective but less toxic therapeutic strategy to block chemotherapy resistance. The inhibition of MAPK pathways, either directly or through Mnks, could be key in blocking the ability of leukemic cells to achieve resistance. In this sense, it has been shown that MNKI-8e, an inhibitor of Mnk, as well as Mnk knockdown generated by short hairpin RNA (shRNA), improves the ability of Ara-C to induce apoptosis in MV4-11 AML cells by suppressing the expression of antiapoptotic proteins such as Mcl-1. Thus, blocking the MAPK pathway through Mnk may be a promising therapeutic strategy to sensitize leukemic cells to Ara-C therapy [55] (Figure 3b(i)).

Abnormal activation of the mTOR pathway is frequently observed in AML patients, especially those with chemoresistance [85] (Figure 3a). The suppression of mTOR is key for the antitumor effect in several tumor models [63,86,87]. For example, emodin (1,3,8-trihydroxy-6-methyl-anthraquinone), a natural anthraquinone extracted from *Rheum officinale* and *Polygonam cuspidatum* with anti-proliferative/proapoptotic effects, suppresses mTOR, Akt and extracellular signal-regulated kinases (ERK) expression. Akt activation indirectly promotes the transcription of antiapoptotic genes, and it directly phosphorylates mTOR [88]. ERK is associated with the development of the multidrug resistance phenotype [89] (Figure 3a). Interestingly, emodin in combination with Ara-C induces proliferation arrest and apoptosis in various leukemic cell lines (U937, HL-60, HL-60/H3, CEM, Jurkat, Molt-4 and CA46) and in HL60/ADR multidrug-resistant cells. In an in vivo leukemia model (HL60/ADR cells inoculated in BALB/c nude mice), high-dose emodin increased the sensitivity to Ara-C, inhibited Bcl-2 expression, inhibited tumor growth and improved survival (Figure 3b(ii)) while being tolerated by the nude mice, without serious adverse effects upon hematological and histopathological examination [65]. Similarly, inhibition of the mTORC1/P70S6K pathway using metformin, a classic hypoglycemic drug for treating patients with type II diabetes, sensitizes AML HL60 cells to Ara-C, producing a synergistic antitumor effect in comparison with Ara-C alone [63] (Figure 3b(iii)).

Another Ara-C sensitization approach focuses on key transcription factors in oncogenic development. cAMP response element binding protein (CREB) is overexpressed in bone marrow cells in 65% of patients with AML and has been associated with an increased risk of relapse [90] (Figure 3a). The deletion of CREB inhibits the proliferation of AML cells without affecting normal hematopoietic activity in murine models of hematopoietic cell transplantation [91]. Niclosamide, which was approved by the FDA 50 years ago for the treatment of *Taenia* infections, has shown therapeutic potential to sensitize leukemic cells to Ara-C, daunorubicin and vincristine. Niclosamide has been shown to inhibit the expression and activity of CREB (Figure 3b(iv)), leading to G1/S phase arrest with subsequent apoptosis, significantly inhibiting the progression of the disease and prolonging survival in a xenograft mouse model [57]. Interestingly, some studies propose that niclosamide has antineoplastic activity in different types of cancer, such as breast, head and neck, kidney and lung cancer, targeting multiple pro-oncogenic signaling pathways, such as NF-κB, Wnt/b-catenin, mTORC1, STAT3 and Notch. In the case of AML, the antineoplastic effect of niclosamide (Figure 3b(iv)) is due to the inhibition of transcription, the binding of NF-κB to DNA, and an increase in ROS levels, which combine to induce apoptosis in leukemic cells [57,92,93,94].

### 4.2. Synergistic Cytotoxicity with Subtoxic Concentrations of Ara-C

Cyclin-dependent kinase (CDK) 4 and CDK6 are frequently overexpressed in human cancers [95] (Figure 3a). Their use as targets in combination with cytotoxic agents has been investigated for their ability to increase cytotoxicity when used in combination therapy [96,97,98]. PD0332991 (palbociclib), a cell-permeable pyrido-pyrimidine with oral availability, has been used as a selective inhibitor of CDK4/CDK6, having a favorable clinical response in the treatment of resistance/refractoriness in patients with CML and breast cancer [56,96,97,98]. Yang et al. studied the in vitro and in vivo effect of PD0332991 as an inhibitor of CDK4/CDK6 in the AML cell line HL-60 and found that it sensitizes leukemic cells to Ara-C through two mechanisms. The first is arrest in the G1 phase and synchronization in the S phase of the cell cycle for a greater incorporation of Ara-C in DNA replication. The second mechanism is increasing the cytotoxic effect of Ara-C by dismantling the homeobox (HOX) A9 oncogene-dependent antiapoptotic pathway—by inhibiting HOXA9, PIM1 transcription is reduced, which activates BAD-dependent apoptosis in tumor cells. The inhibition of CD4/CDK6 by PD0332991 suppressed tumor growth at lower doses of Ara-C in an AML xenograft model. These data suggest that CDK4/CDK6 inhibition sensitizes AML cells to the cytotoxic effect of Ara-C and is capable of killing leukemic cells at reduced Ara-C doses. Thus, PD0332991 may have possible implications for the treatment of AML patients who are unable to tolerate high doses of Ara-C [56] (Figure 3b(v)).

### 4.3. Synergistic Cytotoxicity with Ara-C without Damaging Normal Hematopoietic Tissue

Activated oncogenic Ras in leukemic cells results in constitutive phosphorylation and activation of the MAPK pathway [99], which plays an important role in the regulation of leukemic cell proliferation, leading to a poor prognosis, chemoresistance and disease relapse [55,100] (Figure 3a). Therefore, a therapeutic strategy that can negatively regulate this important pro-oncogenic pathway could be a valuable therapeutic target. Saikia and collaborators showed that heteronemin could increase cell sensitization to Ara-C. Heteronemin is a natural compound of marine origin. In combination with low doses of Ara-C, it produces a much greater synergistic cytotoxic effect than high doses of Ara-C alone. Interestingly, the use of a subtoxic concentration of Ara-C in combination with heteronemin was antileukemic but harmless to cells isolated from the peripheral blood of healthy donors. Molecularly, heteronemin inhibits farnesyl transferase, which is a recognized inducer of Ras activation. Heteronemin also blocks the binding of NF-κB to DNA and significantly reduces the overexpression of c-myc, both of which are widely deregulated processes in leukemic cells. Notably, heteronemin also reduces Ara-C-mediated activation of ERK, JNK and p38, three key molecules involved in MAPK signaling, while almost completely inhibiting AP-1 nuclear translocation, the effector molecule downstream of the MAPK pathway. Thus, the use of heteronemin in a synergistic strategy with Ara-C by inhibiting the farnesylated Ras-MAPK-NF-κB/AP-1 axis has been suggested to sensitize AML cells to Ara-C [66] (Figure 3b(vi)).

The NSP (nucleoside salvage pathway) is essential to obtain the nucleotides necessary for the synthesis of nucleic acids and proteins; other important cellular functions related to the NSP are energy conservation, signaling activity, glycosylation mechanisms and cytoskeletal function [101]. In addition to participating in the transformation of Ara-C to its active form Ara-CTP, dCK is a central enzyme in the NSP pathway. AML cells that acquire chemoresistance by inactivating dCK abolish the NSP, and the only alternative to a sufficient nucleotide supply is the de novo nucleotide synthesis pathway (DNSP) (Figure 3a). The hyperactivation of the mTOR pathway and oncogenes such as MYC, RAS and AKT are key determinants for the activation of the DNSP, which contributes to the proliferation of tumor cells [64,101]. Inhibiting ribonucleotide diphosphate reductase (RNR), a key enzyme in the DNSP pathway, could cut off the metabolic capabilities of tumor cells. Hydroxyurea (HU), an RNR inhibitor used clinically to control myeloproliferative disorders, sickle cell anemia and acquired immunodeficiency syndrome (AIDS), as well as azidothymidine (AZT), a nucleotide thymidine analog drug for treating human immunodeficiency virus (HIV) infection both have the ability to reverse cell resistance to cisplatin in resistant cell lines due to the synergistic activity between HU and AZT based upon their shared deleterious impact on the depletion of cellular dTTP [102]. HU and AZT are interesting candidates as enhancers of Ara-C cytotoxicity in cases of relapse and refractoriness of AML caused by chemoresistance (Figure 3b(vii)). Ara-C-resistant AML sub-cell lines (K562 and Kasumi-1) lacking dCK expression are hypersensitive to the combination of Ara-C with HU plus AZT compared to parental cell lines. This synergistic effect in inhibiting proliferation is also observed in peripheral blood cells from AML patients. In contrast, the combination of HU and AZT does not inhibit the proliferation of nonmalignant cells [64].

## 5. Future Perspectives

Therapeutic strategies against AML based on a single drug have had little impact in increasing the survival of refractory and relapsed patients with multidrug resistance. This literature review has clarified seven strategies that, in combination with Ara-C, have the potential to counteract refractoriness/relapse. The challenge is to show which and how many combinations of the seven options are viable and effective to cure AML, taking special care not to damage normal cells.

We propose the combination of four of these strategies together with Ara-C as a new option to be investigated for its use in AML treatment: emodin + heteronemin + HU + AZT. Combining multiple drugs with Ara-C is not entirely new, as mentioned in the case of fludarabine. The combination of several molecules can promote the apoptosis of leukemic cells in three ways: (1) synergistically increasing the apoptotic effect triggered by Ara-C; (2) inhibiting chemoresistance pathways involving mTOR, Akt, ERK, cMyc, Ras, NFkB and MAPK; and (3) inhibiting the chemoresistance phenotype by decreasing ERK expression and the absence of dCK, which is related to the NSP pathway (Figure 3b). Because these molecules are not toxic in healthy cells, the inclusion of emodin, heteronemin (both of natural origin), HU and AZT would provide a therapeutic option to patients who are not able to tolerate high doses of chemotherapy due to their condition (advanced age, unfavorable karyotype, etc.). This combination is proposed to be evaluated in vitro and in vivo to determine if the use of these molecules in combination can exert an antiproliferative/proapoptotic effect in AML cells and to rule out possible adverse effects arising from their combination.

## 6. Conclusions

Despite the considerable progress that has been made to develop new treatments for patients with AML, the problem of drug resistance continues. High-risk patients and those in relapse currently receive treatments directed against a key target to attack leukemic cells. However, the intrinsic complexity associated with the different MOCs that cells develop before, during and after treatment makes this an almost impossible task. This highlights the need to continue searching for new therapeutic agents, and above all, to develop chemosensitizing strategies capable of limiting and, in the best of cases, blocking the acquisition of resistance to Ara-C, which has been the cornerstone treatment for AML for more than 50 years. This review describes seven strategies that, in combination with Ara-C, have the potential to counteract refractoriness and relapse. The strategy proposed in this work is the combination of different molecules that simultaneously inhibit key MOCs in the development of chemoresistance, making leukemic cells more sensitive to Ara-C. This combination strategy might also synergistically increase the antitumor cell effect of low Ara-C concentrations, with minimal damage to healthy tissue.

## Figures and Tables

**Figure 1 ijms-22-04955-f001:**
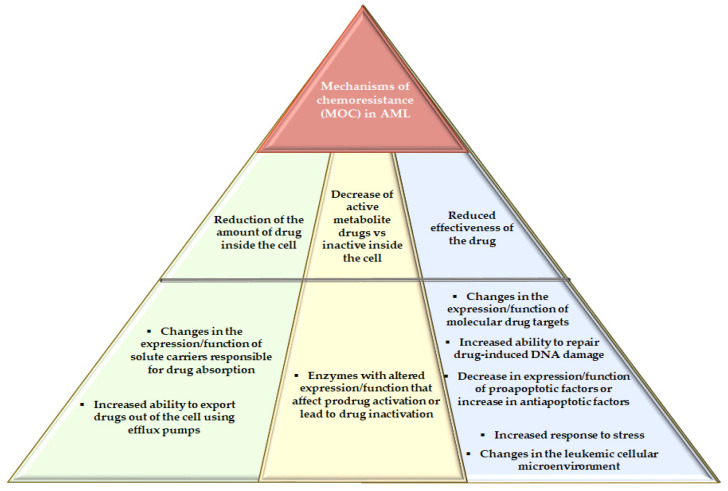
Mechanisms of chemoresistance (MOC) in AML cells.

**Figure 2 ijms-22-04955-f002:**
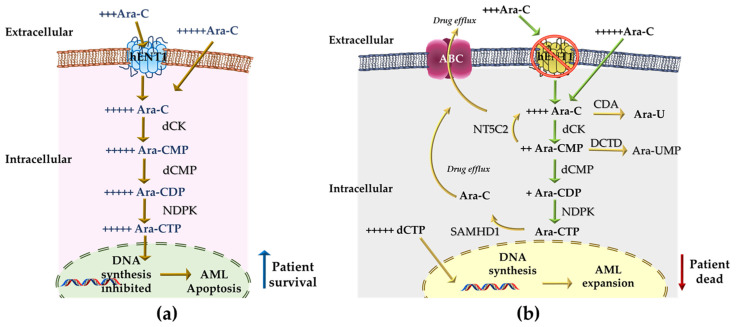
Molecular mechanisms of action and Ara-C resistance. (**a**) Ara-C antineoplastic mechanism of action. Ara-C is transported into the cell by hENT1; at high chemotherapy doses, it enters by passive diffusion. Inside the cell, Ara-C is sequentially phosphorylated by kinases (dCK, dCMP and NDPK) to the active antimetabolite Ara-CTP, which translocates to the nucleus and blocks DNA synthesis. Cell death eventually occurs due to apoptosis, and the patient survives. (**b**) Primary mechanisms of resistance to Ara-C. Either low levels, inactivity or both, of the hENT1 transporter limits the influx of Ara-C. Aberrant expression of ABC transporters increases the expulsion of the drug. A reduced dCK level is the limiting step in the phosphorylation of Ara-C to Ara-CMP; in addition, Ara-CMP can be dephosphorylated back to Ara-C through overexpressed NT5C2, which reduces the availability of Ara-CMP for dCMP. This reduces the concentration of Ara-CDP and eventually Ara-CTP. Furthermore, elevated levels of CDA and DCTD can convert Ara-C and Ara-CMP to the inactive forms Ara-U and Ara-UMP, respectively. SAMDH1 hydrolyzes Ara-CTP back to the inactive form Ara-C, while the increased dCTP concentration (characteristic of AML) competes with the incorporation of Ara-CTP into DNA, further nullifying the effect of chemotherapy. Ara-C: cytarabine; Ara-CMP: Ara-C monophosphate; Ara-CDP: Ara-C diphosphate; Ara-CTP: Ara-C triphosphate; Ara-U: uracil arabinoside; dCMP: deoxycytidine kinase monophosphate; hENT1: human equilibrative nucleoside transporter; dCK: deoxycytidine kinase; CDA: cytidine deaminase; NT5C2: 5’-nucleotidase; DCTD: deoxycytidylate deaminase; SAMHD1: SAM and HD domain containing protein 1; CDP: cytidine diphosphate; NDPK: nucleoside diphosphate kinase; dCDP: deoxycytidine diphosphate; dCTP: deoxycytidine triphosphate.

**Figure 3 ijms-22-04955-f003:**
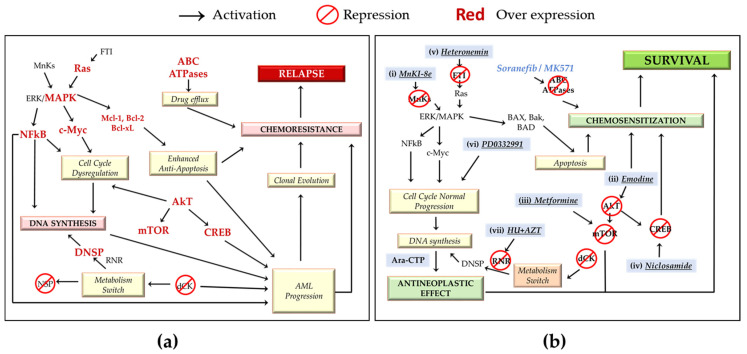
Chemoresistance pathways and chemosensitization in Ara-C chemoresistant cells. (**a**) Ara-C chemoresistance pathways. Leukemic cells develop different MOCs that determine the acquisition of resistance to Ara-C: the overexpression (red and bold) of oncogenes such as RAS, NFkB, MAPK and c-Myc, as well as key modulators of cell function such as AkT and mTOR lead to the activation of oncogenic pathways (CREB overexpression), antiapoptotic factors (Bcl2, Mcl1, Bcl-xL), changes in metabolism (blocking of NSP by dCK depletion and transition to DNSP via the RNP enzyme), and dysregulation of the cell cycle (via overexpression of CDKs and cyclines). This set of MOCs, in addition to the aberrant expression of ABC transporters, is responsible for the cell efflux of chemotherapy drugs and is the cause of chemoresistance and relapse. (**b**) Proposal to induce cytotoxicity in Ara-C chemoresistant AML cells. In the presence of standard doses of Ara-C, the addition of (i) MnKI-8e, (ii) emodin, (iii) metformin or (iv) niclosamide (blue boxes) produces either chemosensitization, a synergistic antitumor effect, apoptosis, blocking mutated pathways (mTOR and CREB), or negative regulation of oncogenes (NFkB, AkT, c-Myc) and key regulators of cell function (MAPK) compared to Ara-C alone, or in combination; (v) heteronemin or (vi) PD0332991 (blue boxes) sensitizes resistant AML cells to reduced or subtoxic concentrations of Ara-C. Standard therapeutic concentrations of Ara-C in the presence of (vi) hydroxyurea and azidothymidine (HU + AZT) (blue boxes) are antileukemic due to the blockade of the RNP enzyme, a key enzyme in the DNSP pathway. Nonmalignant cells are resistant to damage because they have a functional NSP pathway. The use of sorafenib or MK571 efficiently blocks the efflux of Ara-C and Ara-CTP by inhibiting ABC transporters. Some MOCs prevail (metabolic switches), but the therapeutic effect of Ara-C is not compromised.

**Table 1 ijms-22-04955-t001:** Simplified classification of acute myeloid leukemia (AML) based on sensitivity or resistance to chemotherapy.

Type	Characteristics	Treatment Approach	References
Chemosensitive AML at diagnosis	Translocations *RUNX1-RUNXT1* and *CBFB-MYH11*, CBF leukemia (without *KIT* mutation). Diploid AML with *NPM1* or *CEBPα* (without FLT3 mutation). Younger patients without therapy-related AML or antecedent of hematological disorder).	Cytotoxic combination chemotherapy/dose intensification of chemotherapy	[10,11]
Chemoresistant AML at diagnosis	Complex karyotype (≥3 cytogenetic abnormalities) or specific chromosomal aneuploidies (e.g., −5/−5q. −7 and −17/−17p) FLT3-ITD mutation. Older/younger patients with therapy-related AML or antecedent of hematological disorder).	New agents (e.g., molecular targeted or immune-based therapy)	[10,11]
Chemoresistance acquired by clonal evolution	Adaptation to the new environment defined by chemotherapy or new treatment; mutational profile change; survival and proliferation.	Existing treatments (chemotherapy and new agents) without significant increase in survival	[12,13]

**Table 2 ijms-22-04955-t002:** Approved chemotherapies and new therapies in clinical trials for AML.

FDA Approved (Approval Year) or Current Status of Use	Drug Name/Active Ingredient (Clinical Trial Number)	Reference
Main treatment for 50 years in combination with anthracyclines	Cytarabine	[4]
1991	Fludara/fludarabine	[17]
1997	Idamycin PFS/idarubicin	
1997	Etopophos/etoposide	
1998	Cytosar-U/cytarabine	
2004	Vidaza/azacitidine	
2005	Nexavar/sorafenib	
2006	Sutent/sunitinib	
2006	Dacogen/decitabine	
2017	Rydapt/midostaurin	
2017	CPX-351/vyxeos (cytarabine/daunorubicin)	
2017	Idhifa/enasidenib	
2017	Mylotarg/gemtuzumab ozogamicin	
2018	Tibsovo/ivosidenib	
2018	Venclexta/venetoclax	
2018	Xospata/gilteritinib	
Preclinical investigational drugs	NSC-370284, UC-514321, SD36, ALRN-6924, BRD0705, SHP099, MP-A08, BAY 2402234, Meclizine, OG-86, EPZ-6438, Atuveciclib, DB2313, DB2115, DB1976	
Investigational drugs in phase I	BYL719 (NCT01449058), Everolimus (NCT01154439), Pacritinib (NCT02323607), OPB-111077 (NCT03197714), LGH447 (NCT02078609), Sonidegib (NCT02129101), BMS-214662 (NCT00006213), Nintedanib (NCT03513484), PTC299 (NCT03761069), LY2874455 (NCT03125239), Merestinib (NCT03125239), CYC140 (NCT03884829), SEL120 (NCT04021368), Veliparib (NCT00588991), Pevonedistat (NCT03009240), MEK162 (NCT02049801)	
Investigational drugs in phase II	FF-10101-01 (NCT03194685), GSK214795 (NCT01907815), Perifosine (NCT00301938), MK2206 (NCT01253447), Sirolimus (NCT02583893), Temsirolimus (NCT00084916), Glasdegib (NCT03226418), Trametinib (NCT01907815), Selumetinib (NCT00588809), Lenalidomide (NCT00890929), Alisertib (NCT00830518), BI 811283 (NCT00632749), Entospletinib (NCT02343939), Ponatinib Hydrochloride (NCT01620216), Lestaurtinib (NCT00469859), Pexidartini (NCT01349049), JNJ-40346527 (NCT03557970), Semaxanib (NCT00005942), Cediranib Maleate (NCT00475150), Erlotinib Hydrochloride (NCT01664897), GO-203-2c (NCT02204085), Talazoparib (NCT02878785), Selinexor (NCT02835222), HDM201 (NCT03760445)	[18]
Investigational drugs in phase III	Zosuquidar (NCT00046930), Tipifarnib (NCT00093990), Valspodar (NCT00003190)	

**Table 3 ijms-22-04955-t003:** Mechanisms of chemoresistance (MOC) in AML.

Type of Molecular Alteration	Molecule	Reference
Proteins and enzymes	P-gp, MRP1, LRP, GST, TopoII and PKC	[18,36]
Genes	FLT3, WT1, RAS family, MDR1 (ABCB1), SAMHD1, EZH2 and KDM6A have been proposed	[12,36]
miRNAs	Group I, high expression associated with sensitivity: miR-10, miR-27a, let-7a, let-7f, miR-96, miR-128, miR-135a, miR-181a, miR-181b, miR-331 and miR-409. Group II, high expression associated with resistance: miR-20a, miR-32, miR-155, miR-125b, miR-126, miR-210, miR-3151, miR-196b, miR-199a, miR191, miR128, HOTAIR and HOTAIRM1	[33]
Signaling pathways	PI3K/AKT/mTOR, STAT5/PIM, RAS/MAPK, P53, NF-κB, Hh and UPR	[18,36]
Molecules related to drug metabolism	In the case of Ara-C: CDA: irreversibly deaminates Ara-C, changing it to its inactive form, Ara-U SAMHD1: reduces the level of active Ara-CTP through hydrolysis to inactive Ara-C	[37]
Interaction with the tumor microenvironment	SDF-1/CXCR4, FGF2/FGFR1 and VCAM/VLA4 ligand/receptor interaction generates drug resistance similar to FLT3. Hypoxia and acidic pH by maintaining quiescence of leukemic stem cells.	[36]

**Table 4 ijms-22-04955-t004:** Experimental Ara-C sensitization strategies for AML cells.

Ara-C Sensitization Strategies	Molecule	Evidence	Reference
Molecular targets	ABCC4 (MRP4)	ABCC4 protects leukemia cells from Ara-C by through efflux. Inhibiting ABCC4 (e.g., with sorafenib and MK571) or silencing it with an siRNA can reverse Ara-C resistance in AML cells. Abcc4 deficiency in mouse cells sensitizes myeloid progenitors to Ara-C.	[51]
	SAMDH1	SAMDH1 depletion in AML blasts increases sensitivity to Ara-C. Low SAMDH1 expression has been associated with longer survival in a subgroup of patients who received high doses of Ara-C. The combination of high-dose Ara-C with SAMDH1 inhibitors sensitizes cells to chemotherapy.	[52,53,54]
	Mnk	MNKI-8e (an Mnk inhibitor) and an shRNA-generated Mnk knockdown both enhance the ability of Ara-C to induce apoptosis in the human MV4-11 cell line by suppressing MAPK and antiapoptotic proteins.	[55]
	CDK4/6	PD0332991, a CDK4/6 inhibitor, synchronizes HL60 cells in the S phase of the cell cycle, favoring the incorporation of Ara-C at the time of DNA replication, thereby increasing apoptosis. PD0332991 suppressed tumor growth at a lower dose of Ara-C in a xenotransplantation model.	[56]
	CREB	Pretreatment with niclosamide, a CREEB inhibitor, sensitizes HL60 cells to Ara-C, daunorubicin and vincristine, showing a synergistic effect by inhibiting proliferation and reducing the viability of leukemic cells.	[57]
Noncoding RNA	miRNA	miR-23a, miR-21, miR-181b and miR-181 are examples of miRNAs involved in drug resistance and are used to sensitize AML cells to Ara-C. The overexpression of miR-23a decreases the sensitivity to Ara-C, while its knockdown has the opposite effect. Likewise, high miR-23a expression has been correlated with relapse and refractoriness of AML. Downregulating miR-21 significantly sensitizes HL60 cells to Ara-C by inducing apoptosis; this effect is partially due to the upregulation of PDCD4. miR-181b is significantly decreased in human multidrug-resistant leukemia cells and in R/R AML patient samples. The overexpression of miR-181b increases the sensitivity of leukemia cells to doxorubicin and Ara-C and promotes drug-induced apoptosis, at least partially though the direct suppression of its target genes HMGB1 and Mcl-1. In a similar way, miR-181a expression is downregulated in the Ara-C-resistant cell line HL-60/Ara-C compared to the parental cell line HL-60, and overexpression of miR-181a in HL-60/Ara-C cells sensitizes the cells to Ara-C treatment by inducing apoptosis.	[33,58,59,60]
Epigenetic regulation	MTF2–MDM2	MTF2 deficiency is related to drug resistance and refractoriness in AML. MTF2 upregulation or MDM2 inhibition sensitizes cells from AML patients to treatment with Ara-C and daunorubicin.	[61]
	Remodeler CHD4	CHD4 depletion in U937, MV4-11 and AML-3 cell lines and in primary AML cells sensitizes them to treatment with Ara-C and daunorubicin by relaxing chromatin and impairing the ability to repair the double-stranded DNA.	[62]
	KDM6A	The downregulation of KDM6 favors drug resistance in K562 and MM-6 cell lines and is related to decreased ENT1 expression. The restoration of KDM6A expression in KDM6A-null cells of the TPH-1 and K562 KDM6A KO lines suppresses proliferation, and the cells are sensitized to Ara-C.	[39]
Nonspecific substances	Metformin	Metformin sensitizes leukemic cells to Ara-C treatment by inhibiting the mTORC1/P70S6K pathway, thereby promoting apoptosis. In vivo, in leukemic cell transplants in nude mice, the combination of metformin and Ara-C produces a synergistic antitumor effect compared to the use of Ara-C alone.	[63]
	Hydroxyurea and azidothymidine	In sub-cell lines resistant to Ara-C and in peripheral blood cells from patients with AML, treatment with HU and AZT in combination with Ara-C results in a synergistic effect to inhibit cell growth.	[64]
	Emodin	In combination with Ara-C, emodin inhibits proliferation and promotes apoptosis in leukemic cell lines, including HL60/ADR. In vivo, the administration of high doses of emodin increases sensitivity to Ara-C, inhibiting tumor growth and improving survival.	[65]
	Heteronemin	The combination of heteronemin and low-dose Ara-C produces an improved synergistic cytotoxic effect towards AML cells compared with high-dose Ara-C alone.	[66]

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
