# Peer review of "Inhibitors of Chemoresistance Pathways in Combination with Ara-C to Overcome Multidrug Resistance in AML. A Mini Review"

_ijms, 2021, doi:10.3390/ijms22094955_

Round 1

Reviewer 1 Report

The manuscript is focused on an interesting therapeutic issue in onco-hematology, is well developed, brings together an impressive amount of informations and reads quite well. While the pre-clinical laboratory part is sufficiently well prepared, some work is needed to improve the links with clinical world, in order to make this type of studies more attractive to the clinicians.

CRITICISMS

  1. In the Introduction the Authors should clearly state that the mainstay of standard AML therapy are cytarabine and anthracyclines, and that based on historical experience less than 50% of all AML patients are cured by this initial approach followed by any type of consolidation and/or HCT. An excellent recent reference to add to this point is Kantarjian et al, Blood Cancer J 2021.
  2. Next, it should equally clearly state that while an improvemt is expected with new treatment modalities, undertanding and modifying resistance mechanisms to currently available drugs may be another useful way to cope with the problems. These are the driving concepts of the review, to capture the readers interest.
  3. Paragraph 2. The FAB classification can be safely omitted (lines 68-77).
  4. One major point is how to identified drug-resistant patients for experimental therapies. While WHO/ELN/genetic classifications provide a general prognostic framework, a more refined approach for personalized medicine would involve the nalysis of predictive markers of resistance. These are the persistence of MRD post-induction and ex vivo drug sensitivity testing assays. These two issues should be added and adequately developed.
  5. Again of great interest to clinicians: in the past fludarabine was added to cytarabine as an agent able to partially by-pass cytarabine resistance. Therefore that was the first ever attempt to overcome cytarabine resistance. Indeed fludarabine-based regimens appeared relatively effective in R/R AML and appeared superior to standard regimens in some upfront trials. This experience is supporting your proposals and should be quoted. Moreover you may comment on the limitations of fludarabine as cytarabine sensitizer compared to the new agents you are reviewing here.
  6. Several sections with poor grammar and style: extensive revision required.

Reviewer 2 Report

Summary: This review article summarizes what is known about mechanisms of cytarabine (Ara-C) resistance in acute myeloid leukemia (AML). It is comprehensive and nicely written, but would benefit from some minor changes outlined below.

Comments:

  1. Table 1 is difficult to read with the text justified to the center. Please justify the text to the left. Under the characteristics column, please separate the different characteristics onto separate lines in the table, not just separate sentences.
  2. Lines 98-104: The sentence is very long and should be broken up into two sentences.
  3. Table 2 could be improved by providing the references to each treatment option listed. If a publication is not yet available, please including the clinical trial number associated with the trial.
  4. Line 193: MDR stands for multidrug resistance, not resistance multidrug.
  5. Figure 1, center label: Decrease of active agents vs inactive inside the cell. Could the authors please clarify what you mean here? At first I thought you were talking about activation of oncogenes or inactivation of tumor suppressors. Perhaps a new label would be helpful.
  6. In Table 3 under microRNAs, there no molecules listed in the second column, it only says ‘DNA damage, cell cycle aberration, apoptosis, and cell death.’ These are not molecules. Perhaps you should instead list some specific microRNAs that have been implicated in therapy resistance of AML.
  7. Table 4, 1st column: It is currently labeled ‘Experimental strategies kind.’ Please revise to clarify.
  8. Table 4: miR-23 is not the only non-coding RNA implicated in AML therapy resistance. Please expand this to include more.
  9. Lines 293-300: Very long sentence. Please break it up into 2 or 3 sentences.
  10. Please change synergic to synergistic throughout the document.
  11. Figure 3 is very confusing. Please consider revising so that the reader understands the meaning behind the figure. For instance, upon first glance, it seems like repressing MCL-1 or BCL-2 leads to chemoresistance, but we know that is not the case.
  12. Line 387: Should CDK5 be CDK6?
  13. Lines 422-431: Another very long sentence. Please break it up to be more clear and concise.
  14. There is no talk about the role of leukemia stem cells or aberrant oxygen metabolism and how they contribute to therapy resistance. The paper would benefit from an additional paragraph summarizing these recent findings, and how they are being targeted in the clinic (e.g. venetoclax with 5-azacitidine).

Round 2

Reviewer 1 Report

Nothing to add